# Chiral superconductivity in UTe₂ probed by anisotropic low-energy excitations

Kota Ishihara [1] ✉, Masaki Roppongi[1], Masayuki Kobayashi[1], Kumpei Imamura[1], Yuta Mizukami[1,3], Hironori Sakai [2], Petr Opletal[2], Yoshifumi Tokiwa[2], Yoshinori Haga [2], Kenichiro Hashimoto [1] & Takasada Shibauchi [1] ✉

Chiral spin-triplet superconductivity is a topologically nontrivial pairing state with broken time-reversal symmetry, which can host Majorana quasiparticles. The heavy-fermion superconductor UTe₂ exhibits peculiar properties of spin-triplet pairing, and the possible chiral state has been actively discussed. However, the symmetry and nodal structure of its order parameter in the bulk, which determine the Majorana surface states, remains controversial. Here we focus on the number and positions of superconducting gap nodes in the ground state of UTe₂. Our magnetic penetration depth measurements for three field orientations in three crystals all show the power-law temperature dependence with exponents close to 2, which excludes single-component spin-triplet states. The anisotropy of low-energy quasiparticle excitations indicates multiple point nodes near the $k_y$- and $k_z$-axes in momentum space. These results can be consistently explained by a chiral $B_{3u} + iA_u$ non-unitary state, providing fundamentals of the topological properties in UTe₂.

Since the discovery of superconductivity in the nonmagnetic uranium-based compound UTe₂, the nature of its superconducting state has been extensively studied[1–31]. Recent studies have reported several anomalous superconducting properties, including extremely high upper critical field much beyond the Pauli limit[1–3], reentrant superconductivity[3,4], and little reduction of the Knight shift in the nuclear magnetic resonance (NMR)[1,5]. These results suggest odd-parity spin-triplet pairing in UTe₂, as in the case of uranium-based ferromagnetic superconductors[32]. The symmetry of the superconducting order parameter is closely related to the superconducting gap structure, and the previous studies of low-energy quasiparticle excitations, such as specific heat, thermal transport, and magnetic penetration depth measurements[6–8], support the presence of point nodes, consistent with the spin-triplet pairing states.

More intriguingly, recent scanning tunneling microscopy[9], optical Kerr effect[10], and microwave surface impedance measurements[6] suggest time-reversal symmetry breaking (TRSB) in the superconducting state at ambient pressure. As the sign of imaginary part changes under the time-reversal transformation, the chiral TRSB state requires multiple order parameter components in complex form. We note that high-pressure studies reveal several superconducting phases[11–13], suggesting the presence of multiple order parameters under pressure. Thus, UTe₂ is a prime candidate of a topological chiral spin-triplet superconductor. However, the symmetry of the odd-parity vector order parameter **d**, whose magnitude is the gap size and whose direction is perpendicular to the spins of Cooper pairs, is still highly controversial. Especially, the nodal structure of order parameter and whether or not it is chiral in the ground state are important issues to understand the possible topological properties of UTe₂.

The crystal structure of UTe₂ (Fig. 1a) is classified into the point group $D_{2h}$, whose irreducible representations (IRs) of odd-parity order parameters are listed in Table 1. In the cases of $B_{1u}$, $B_{2u}$, and $B_{3u}$ states, point nodes in the superconducting gap function exist on the $k_z$-, $k_y$-, and $k_x$-axes (Fig. 1b), respectively, while the $A_u$ state is fully gapped. When the odd-parity order parameter **d** is represented by a single IR, the positions of nodes can be detected by the temperature dependence of the change in magnetic penetration depth, $\Delta\lambda(T) \equiv \lambda(T) - \lambda(0)$. This is because the low-temperature superfluid density $\lambda^{-2}(T)$, which is

[1]Department of Advanced Materials Science, University of Tokyo, Kashiwa, Chiba 277-8561, Japan. [2]Advanced Science Research Center, Japan Atomic Energy Agency, Tokai, Ibaraki 319-1195, Japan. [3]Present address: Department of Physics, Graduate School of Science, Tohoku University, 6-3, Aramaki Aza-Aoba, Aoba-ku, Sendai 980-8578, Japan. ✉e-mail: k.ishihara@edu.k.u-tokyo.ac.jp; shibauchi@k.u-tokyo.ac.jp

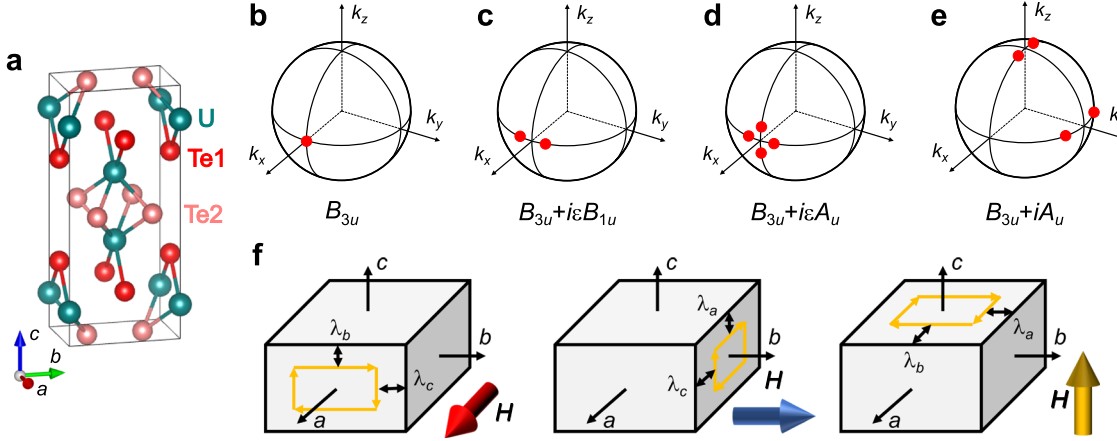

**Fig. 1 | Nodal positions of several pairing states in UTe$_2$ and geometries of anisotropic penetration depth measurements. a** Crystal structure of UTe$_2$. Positions of the point nodes (red points) for $B_{3u}$ (**b**), $B_{3u} + i\varepsilon B_{1u}$ (**c**), $B_{3u} + i\varepsilon A_u$ (**d**), and $B_{3u} + iA_u$ (**e**) order parameters, where $\varepsilon$ is a sufficiently small real number. **f** Schematic relations between the directions of the ac magnetic field (big arrows) and $\lambda$ components.

determined by thermally-excited quasiparticles near the nodes, depends strongly on the directions of the shielding supercurrent density $\mathbf{j}_s$ and the point nodes (whose direction is defined as $\mathbf{l}$). As a result, when the point nodes are directed along crystallographic $\alpha$ axis ($\mathbf{l} \| \alpha$), $\Delta\lambda_\alpha(T)$ follows $T^2$ dependence, while for perpendicular axes $\beta$ and $\gamma$, $\Delta\lambda_\beta(T)$ and $\Delta\lambda_\gamma(T)$ should follow $T^4$ dependence (Table 1)[33]. Here, the subscript $i$ of $\lambda_i$ represents the direction of supercurrent density $\mathbf{j}_s$.

On the other hand, when two symmetries in different IRs accidentally admix to form a TRSB complex order parameter, point nodes are generally located away from the high symmetry axes, and various nodal structures become possible[10,14]. Considering the $B_{3u} + i\varepsilon B_{1u}$ ($B_{3u} + i\varepsilon A_u$) state, for example, where $\varepsilon$ is a sufficiently small real number, a point node of the $B_{3u}$ state splits into two (four) point nodes as depicted in Fig. 1c (Fig. 1d) (for more details, see Supplementary Information I). These split point nodes can be identified as topological Weyl nodes defined by a Chern number[10,14], and corresponding Majorana arc surface states are expected[34]. Although an experimental determination of the exact positions of the nodes is quite challenging in these cases, we can summarize expected nodal positions for different relative sizes of two components in the complex order parameters in Table 2. Thus, by detecting the anisotropy of quasiparticle excitations through direction-dependent physical quantities, such as $\Delta\lambda_i(T)$, we can pin down the superconducting symmetry among the non-chiral and chiral states listed in Tables 1 and 2, respectively.

We use three independent measurements of resonant frequency of the tunnel-diode oscillator (see Methods) with weak ac magnetic field along the $a$-, $b$-, and $c$-axes, in which the shielding current flows perpendicular to the field as described in Fig. 1f. Thus the frequency shift $\Delta f(T)$ consists of two penetration depth components perpendicular to the field direction. As a result, in the single component order parameter cases for point nodes along the $\alpha$ direction, $\Delta f(T)$ for $H_\omega \| \alpha$ is the sum of $\Delta\lambda_\beta(T)$ and $\Delta\lambda_\gamma(T)$ components and thus follows $T^4$ dependence, while $\Delta f(T)$ for $H_\omega \perp \alpha$ should follow $T^2$ dependence at low $T$. We

stress that in our measurements the sample is in the Meissner state. Therefore, our approach is an ideal way to investigate the superconducting symmetry in the ground state in the zero-field limit at ambient pressure.

## Results

### Specific heat
Figure 2a-c show $\Delta f(T)$ in three single crystals of UTe$_2$ denoted as #A1, #B1, and #C1 respectively (see Supplementary Information V). We observe a large change in $\Delta f(T)$ at 2.1 K (#A1), 1.75 K (#B1), and 1.65 K (#C1) corresponding to the superconducting transition. We note that, while crystals #B1 and #C1 are grown by chemical vapor transport (CVT) method, crystal #A1 is grown by molten salt flux (MSF) method and the transition temperature is the highest value ever reported[15]. The clear superconducting transition at $T_c = 2.1$ K, 1.75 K, and 1.65 K for each sample, is also reproduced in the specific heat data (Fig. 2d). Here, we emphasize that a single jump clearly seen in crystals #A1 and #B1 do not necessarily contradict the multi-component order parameter discussed later, because in the Landau theory the jump heights have nontrivial dependence on the coefficients of the fourth power terms of the free energy for chiral superconducting order parameters[11,35]. However, a recent study[16] has reported that the presence or absence of the double transitions depends strongly on the crystal growth conditions, and the origin of the double transitions is still highly controversial[17,18]. Figure 2e shows the low-temperature electronic specific heat $C_e/T$ in crystal #A1 as a function of $(T/T_c)^2$. Here, the subtracted phonon contribution is estimated from the previous measurements[19]. We can find large quasiparticle excitations following $C_e/T \propto (T/T_c)^2$ down to $0.12T_c$, which is an expected behavior in a

## Table 1 | Basis functions, nodal types, and temperature dependence of the magnetic penetration depth for odd-parity order parameters in the point group $D_{2h}$

| IR | Basis functions | Nodes | $\Delta\lambda(T)$ |
|---|---|---|---|
| $A_u$ | $k_x\hat{x}, k_y\hat{y}, k_z\hat{z}$ | None | $\Delta\lambda_{a,b,c} \propto \exp(-\|\mathbf{d}\|/T)$ |
| $B_{1u}$ | $k_y\hat{x}, k_x\hat{y}, k_xk_yk_z\hat{z}$ | Point ($k_z$) | $\Delta\lambda_{a,b} \propto T^4, \Delta\lambda_c \propto T^2$ |
| $B_{2u}$ | $k_z\hat{x}, k_xk_yk_z\hat{y}, k_x\hat{z}$ | Point ($k_y$) | $\Delta\lambda_{c,a} \propto T^4, \Delta\lambda_b \propto T^2$ |
| $B_{3u}$ | $k_xk_yk_z\hat{x}, k_z\hat{y}, k_y\hat{z}$ | Point ($k_x$) | $\Delta\lambda_{b,c} \propto T^4, \Delta\lambda_a \propto T^2$ |

## Table 2 | Expected positions of point nodes in chiral superconducting states with the order parameter d = d$_1$ + id$_2$, where d$_1$ and d$_2$ are classified into different IRs

| IR | | Positions of point nodes | | |
|---|---|---|---|---|
| $d_1$ | $d_2$ | $\|d_1\| \ll \|d_2\|$ | $\|d_1\| \approx \|d_2\|$ | $\|d_1\| \gg \|d_2\|$ |
| $B_{1u}$ | $B_{2u}$ | near $k_y$-axis | near $k_x$-axis | near $k_z$-axis |
| $B_{2u}$ | $B_{3u}$ | near $k_x$-axis | near $k_z$-axis | near $k_y$-axis |
| $B_{3u}$ | $B_{1u}$ | near $k_z$-axis | near $k_y$-axis | near $k_x$-axis |
| $B_{1u}$ | $A_u$ | None | near $k_x$- and/or $k_y$-axes | near $k_z$-axis |
| $B_{2u}$ | $A_u$ | None | near $k_z$- and/or $k_x$-axes | near $k_y$-axis |
| $B_{3u}$ | $A_u$ | None | near $k_y$- and/or $k_z$-axes | near $k_x$-axis |

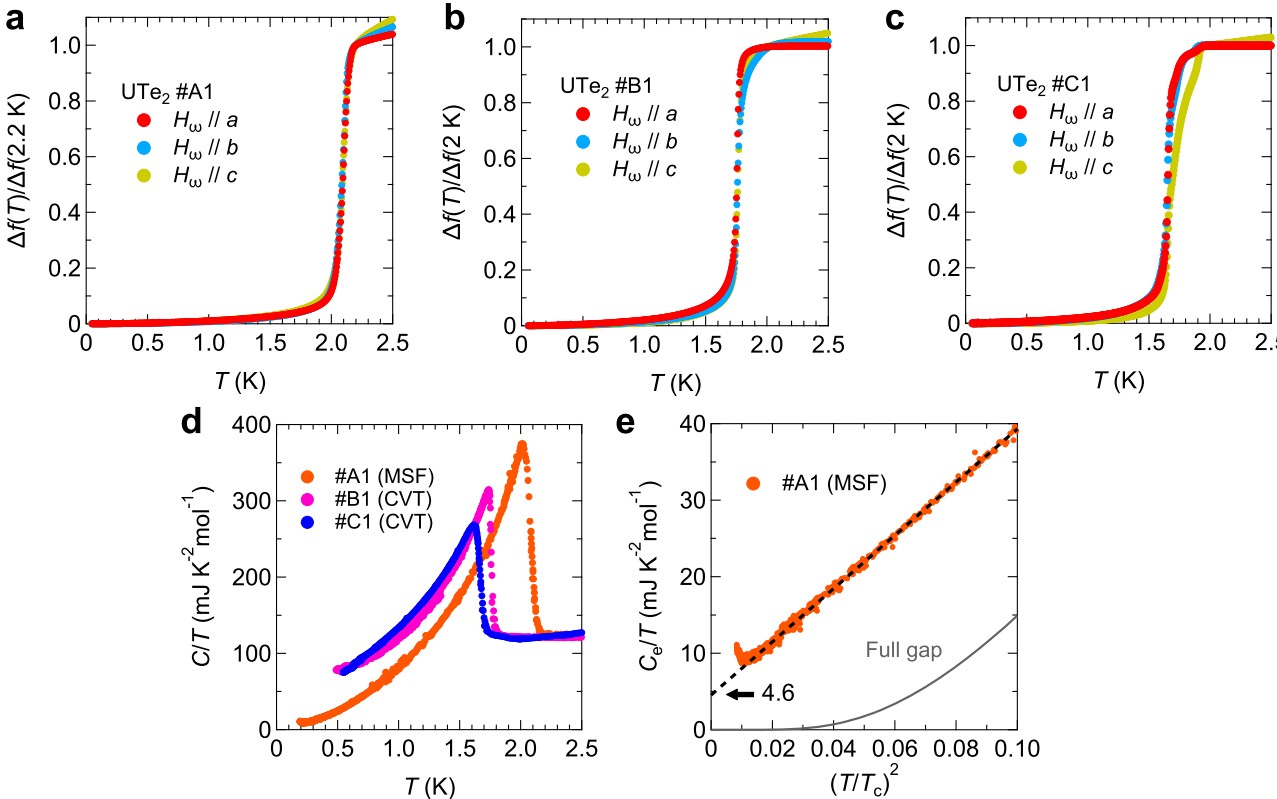

**Fig. 2 | Superconducting transitions in UTe₂. a-c**, Overall temperature dependence of the frequency shift $\Delta f$ in UTe₂ for crystal #A1 (**a**), #B1 (**b**), and #C1 (**c**). The data of #A1 (#B1 and #C1) are normalized by the value at 2.2 K (2 K). **d**, Temperature dependence of the specific heat $C$ divided by $T$ in crystals #A1, #B1, and #C1 measured at zero field. **e**, Enlarged views of the electronic specific heat $C_e/T$ in crystal #A1 at low temperatures as a function of $(T/T_c)^2$. The black broken line represents the linear fitting of the data. The gray solid line is the theoretical curve expected for a full gap structure with the gap size $\Delta = 2.2k_BT_c$, which shows a similar size of the specific heat jump with the experiments.

point-nodal superconductor. Furthermore, the residual electronic specific heat $\gamma_0$ estimated from the extrapolation of the $C_e/T \propto (T/T_c)^2$ relation is 4.6 mJK⁻²mol⁻¹, leading to $\gamma_0/\gamma_n = 0.038$, where $\gamma_n$ is the Sommerfeld coefficient in the normal state. The small $\gamma_0/\gamma_n$ value confirms the high quality of crystal #A1. We note that the upturn in $C_e/T$ observed below $0.12T_c$ is caused by the nuclear Schottky contribution[15]. While the observed $C_e/T \propto (T/T_c)^2$ behavior and small $\gamma_0/\gamma_n$ value indicate the presence of point nodes in the gap structure, we cannot discuss the superconducting symmetry and position of point nodes from the specific heat. Therefore, we rather focus on the low-temperature penetration depth data.

**Magnetic penetration depth**

The key results are the temperature dependence of $\Delta f(T)$ at low $T$, which is shown for crystals #A1, #B1, and #C1 in Fig. 3a-c, respectively. The black solid lines represent fitting curves for the data below $0.3T_c$ using the power-law function, $\Delta f(T) \propto T^n$. We note that the exponent values $n = 2.11$ (#A1), 2.07 (#B1), and 1.95 (#C1) for $H_\omega \| c$ are consistent with the previous $ab$-plane penetration depth studies[6,7]. From the fittings, we find that the obtained exponent values for all field directions are nearly equal to 2 or less than 2 in all the samples. This feature can be more clearly seen by plotting $\Delta f(T)$ as a function of $(T/T_c)^2$ as shown in Fig. 3d-f, where all data show almost linear or convex downward curvatures at low $T$. Thus, our results contradict any cases of single-component odd-parity order parameters, in which $\Delta f(T)$ should follow $T^4$ dependence when the applied field is directed to the point node direction. Another feature of our data is that the exponent values obtained from $\Delta f(T)$ for $H_\omega \| a$ and $H_\omega \| b$ are smaller than that for $H_\omega \| c$ in crystals #B1 and #C1. Considering that $\Delta f(T)$ consists of two $\Delta \lambda_i(T)$ components perpendicular to the magnetic field (Fig. 1f), our

exponent analysis on these samples indicates that the exponent value of $\Delta \lambda_c(T)$ is smaller than those of $\Delta \lambda_a(T)$ and $\Delta \lambda_b(T)$, which will be discussed in more detail below.

For further investigations of the gap structure, we extract $\Delta \lambda_i(T)$ separately from the $\Delta f(T)$ data for three different field orientations, by considering the geometry of the sample (see Supplementary Information VI). Such an analysis is valid when the magnetic penetration depth is much shorter than the sample dimensions, which holds at low temperatures. To compare the quasiparticle excitations along each crystallographic axis, we discuss the normalized superfluid density $\rho_{s,i}(T) = \lambda_i^2(0)/\lambda_i^2(T)$ and normalized penetration depth $\Delta \lambda_i(T)/\lambda_i(0)$ for three supercurrent directions $i = a$, $b$, and $c$, in which evaluations of $\lambda_i(0)$ values are needed. The anisotropy of $\lambda(0)$ can be estimated by the anisotropy of coherence length $\xi$ which can be determined from the initial slope of the temperature dependence of upper critical field $H_{c2}(T)$, when for simplicity we ignore the anisotropy of gap function (see Supplementary Information X). From the $H_{c2}(T)$ data of an ultra-clean sample[20], we estimate $\lambda_a(0) : \lambda_b(0) : \lambda_c(0) = \xi_a^{-1}(T_c) : \xi_b^{-1}(T_c) : \xi_c^{-1}(T_c) = 2.01 : 1 : 3.90$. By using the value $\sqrt{\lambda_a(0)\lambda_b(0)} \approx 1$ μm estimated from the previous penetration depth studies[6,7], we obtain $\lambda_a(0) = 1420$ nm, $\lambda_b(0) = 710$ nm, and $\lambda_c(0) = 2750$ nm.

The obtained normalized $\Delta \lambda_i(T)/\lambda_i(0)$ as a function of $T/T_c$ for three directions of crystals #A1, #B1, and #C1 are shown in Fig. 4a-c, respectively. First of all, the exponent values $n_i$ obtained from the power-law fitting in $\Delta \lambda_i(T)/\lambda_i(0)$ data are all nearly equal to 2 or less than 2, which is again inconsistent with all the cases of the single component order parameter. Especially, as expected from $\Delta f(T)$ data, $n_c = 1.60$ in crystal #C1 and $n_c = 1.84$ in crystal #B1 are smaller than $n_a \approx n_b \approx 2$. The relatively small $n_c$ in crystals #B1 and #C1 can be more clearly seen by plotting the data as a function of $(T/T_c)^2$ (Fig. 4b,c,

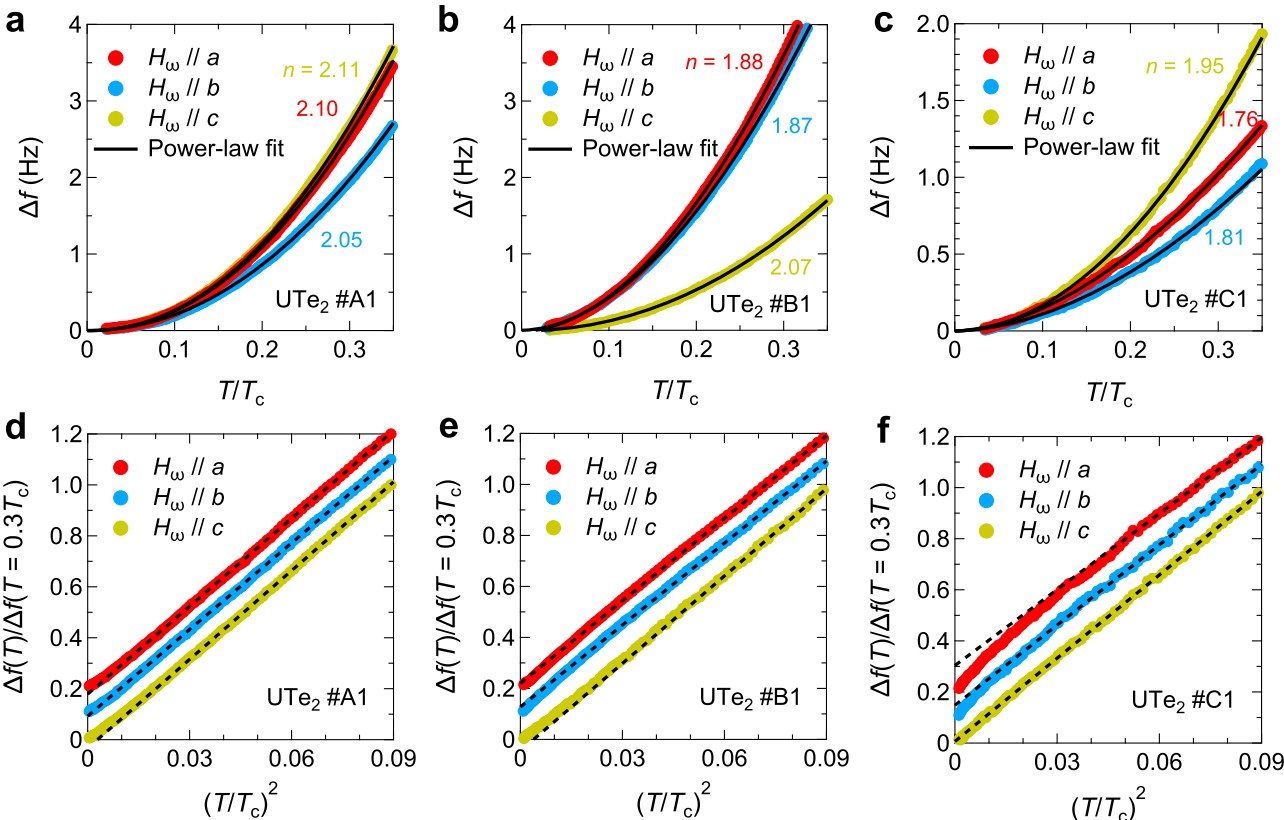

**Fig. 3 | Anisotropic frequency shift in UTe$_2$ for three field directions.** Low-$T$ behavior of $\Delta f$ in UTe$_2$ for crystal #A1 (**a**), #B1 (**b**), and #C1 (**c**) as a function of $T$ normalized by $T_c$. Solid lines represent the fitting curves with the power-law function. $\Delta f(T)$ normalized by the value at $T = 0.3T_c$ as a function of $(T/T_c)^2$ in UTe$_2$ for crystals #A1 (**d**), #B1 (**e**), and #C1 (**f**). Dashed lines represent $T^2$ dependence. The data with the field along the $a$- and $b$-axes are vertically shifted by 0.2 and 0.1, respectively.

inset). We note that the above discussions are independent from the fitting range of the power-law dependence of $\Delta\lambda(T) \propto T^n$ (see Supplementary Information VII). Another consequence of $\Delta\lambda(T)/\lambda(0)$ results is that the quasiparticle excitations along the $b$- and $c$-axes are much larger than those along the $a$-axis, implying a highly anisotropic nodal structure. Figure 4d-f show the normalized superfluid density, $\rho_{s,i} \equiv \lambda_i^2(T)/\lambda_i^2(0)$, along each crystallographic axis for crystals #A1, #B1, and #C1, respectively, plotted against $T/T_c$. Compared with theoretical curves for the single order parameter with the supercurrent density $\mathbf{j}_s$ parallel and perpendicular to the direction of point nodes $\mathbf{I}$, the amount of the excitations along the $a$-axis is clearly smaller than the $\mathbf{j}_s\|\mathbf{I}$ case, while those along the $b$- and $c$-axes are as large as the $\mathbf{j}_s\|\mathbf{I}$ case.

## Discussion

Having established that our anisotropic superfluid density data exclude the single-component odd-parity order parameters, we now discuss the superconducting gap structure based on our experimental results. Adding two order parameters with preserving time-reversal symmetry will not split the point nodes and cannot account for our results (see Supplementary Information II), and thus we need to consider chiral superconducting states formed by two order parameters in different IRs (Table 2). First, we consider chiral superconducting states formed by two $B_u$ IRs. As shown in Supplementary Information I, these chiral states have point nodes located on a symmetric plane of the momentum space. Therefore, small quasiparticle excitations and an exponent value $n_i > 2$ of $\Delta\lambda_i(T)$ are expected for the direction perpendicular to the plane. However, experiments show $n_i \lesssim 2$ for all directions, suggesting that these chiral states are unlikely to be realized in UTe$_2$. Thus, we focus on the chiral superconducting states consisting

of the $A_u$ and a $B_u$ IRs. Considering the observed large quasiparticle excitations along the $b$- and $c$-axes, we conclude that the $B_{3u} + iA_u$ pairing state is most consistent with our experiments (see Table 2). The reason is that for the $B_{3u} + iA_u$ state with similar sizes of $|\mathbf{d}_{B3u}|$ and $|\mathbf{d}_{Au}|$ components, multiple point nodes can exist near the $k_y$- and $k_z$-axes, leading to larger excitations along the $b$- and $c$-axes than along the $a$-axis. Thus, the quicker decrease of our $\rho_{s,b}$ and $\rho_{s,c}$ data than the $\rho_{s,a}$ data (Fig. 4d-f) is consistent with the $B_{3u} + iA_u$ pairing state. This state can be supported by a recent theoretical study based on the periodic Anderson model, which suggests almost equally stable $B_{3u}$ and $A_u$ states at ambient pressure[21]. We note that the chiral $B_{3u} + iA_u$ state is non-unitary with finite $\mathbf{d} \times \mathbf{d}^*$ (see Supplementary Information IV), and for the system close to a ferromagnetic quantum critical point, theory shows that such a non-unitary complex order parameter may become stable[22]. Moreover, recent studies of NMR Knight shift[5] suggest finite $\hat{y}$ and $\hat{z}$ components of $\mathbf{d}$, which is consistent with the $B_{3u} + iA_u$ state. However, we note that the anisotropies of $\lambda_i(0)$ and $\xi_i^{-1}$ are not completely the same in the case of anisotropic superconductors (Supplementary Information X). To confirm the chiral $B_{3u} + iA_u$ state, more direct measurements of $\lambda_i(0)$ are highly desired.

Next we discuss the sample dependence of the small exponent value $n_c$. We found that the $n_c$ value systematically approach 2 as $T_c$ gets higher (Fig. 4a-c), suggesting that impurity effect is related to the $n_c$ value. In the line node case, the exponent value larger than the clean limit $n = 1$ and smaller than $n = 2$ can be interpreted as a consequence of nonmagnetic impurity scatterings[36], quantum criticality[37], or non-local effects[38]. However, these possibilities can be excluded in the case of UTe$_2$ (see Supplementary Information III). In the presence of point nodes in the gap structure, the impurity scattering can affect only the amplitude of the low-energy excitations, but does not symply change

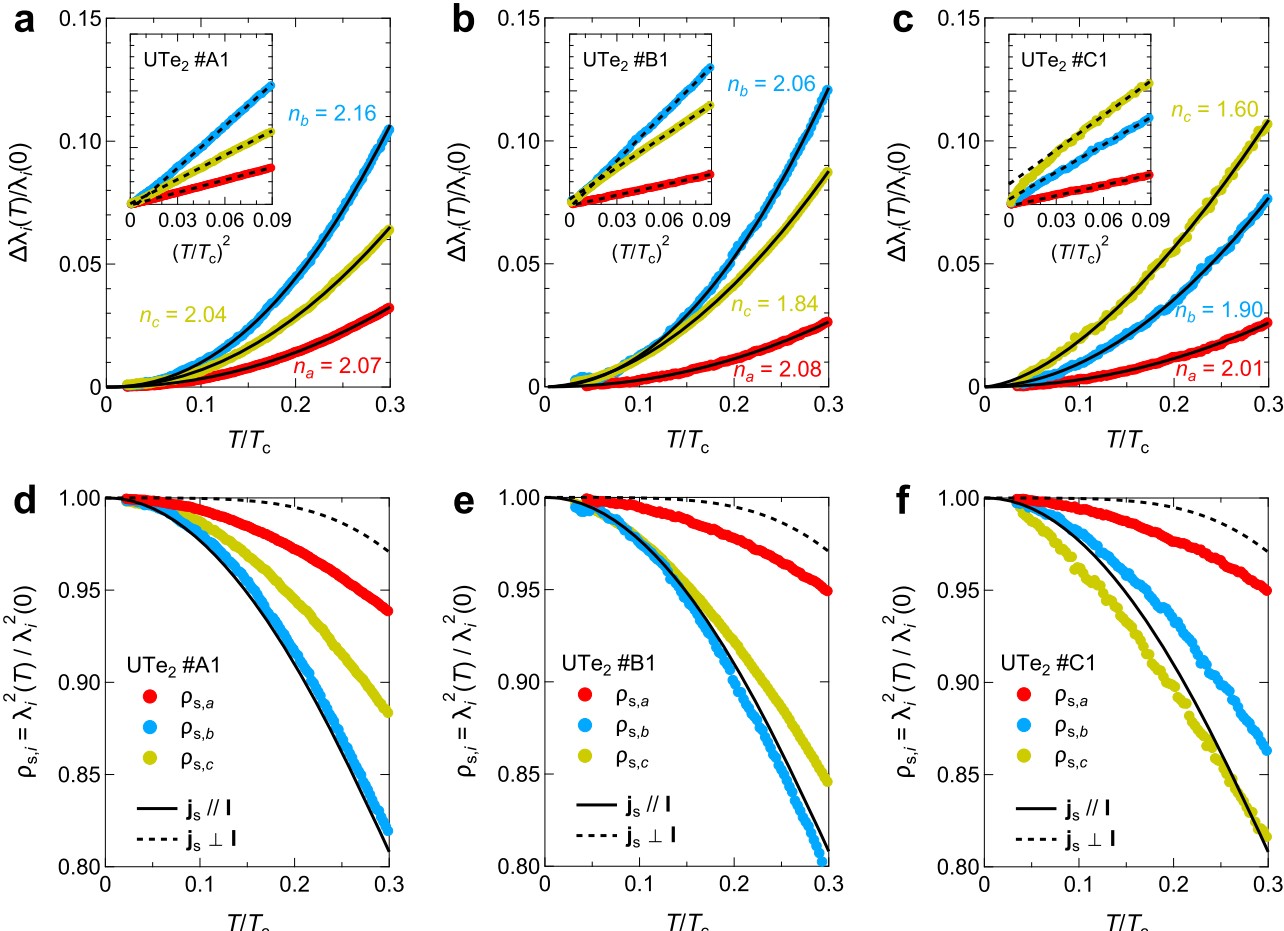

**Fig. 4 | Anisotropic penetration depth in UTe₂ for three supercurrent directions.** $\Delta\lambda_i(T)$ normalized by $\lambda_i(0)$ for $i = a$, $b$, and $c$ as a function of $T/T_c$ for crystals #A1 (**a**), #B1 (**b**), and #C1 (**c**). Solid lines in **a-c** represent the fitting curves of the power-law function and the obtained exponent values are shown in the figures. The inset shows the same data plotted against $(T/T_c)^2$, and the black dashed lines represent $T^2$ dependence. Normalized superfluid density calculated from $\Delta\lambda_i(T)/\lambda_i(0)$ data for crystals #A1 (**d**), #B1 (**e**), and #C1 (**f**). Solid and dashed lines represent the theoretical curves for the single order parameter case for $j_s$ parallel and perpendicular to the nodal direction I, respectively.

the exponent value of $\Delta\lambda(T) \propto T^2$ [6,33]. Here we propose an interference effect of the two point nodes located closely as the origin of the sample-dependent $n_c$ values. In the $B_{3u} + iA_u$ state, the main contributions to $\Delta\lambda_c(T)/\lambda_c(0)$ come from two pairs of two point nodes near the north and south poles along the $k_z$-axis (Fig. 1e). When the distance of the two nodes near the pole gets sufficiently short, the low-energy excitations can no longer be treated as a sum of the contributions from two independent point nodes, and this interference effect can lead to an exponent value less than 2. We have confirmed that a simple model based on the $B_{3u} + iA_u$ state can indeed lead to an exponent near the experimental $n_c$ value when the two point nodes are sufficiently close to each other (see Supplementary Figs. S12 and S13). Moreover, the observed trend that $n_c$ gets smaller for samples with lower $T_c$ can be consistently explained if we consider that the anisotropic $B_{3u}$ component is more sensitive to impurity scattering than the isotropic $A_u$ component. The reduced $B_{3u}$ component in disordered samples would move the point nodes near the $k_z$ axis closer to each other (Supplementary Information IX), making the interference effect more prominent as observed. We note, however, that the detailed impurity effect on the chiral state requires more microscopic understanding, which deserves further studies.

We should mention that, while recent quantum oscillation measurements revealed quasi-two-dimensional Fermi surfaces (FSs) [23], their precise shape is still under debate. Some theoretical and ARPES studies suggest the presence of a heavy three-dimensional FS around

the $Z$ point, in addition to quasi-two-dimensional FSs mainly formed by U $d$ orbitals and Te $p$ orbitals [14,21,24,25]. The presence of the three-dimensional FS is also supported by the relatively isotropic transport properties [26]. These indicate that there are FSs near $k_x$-, $k_y$-, and $k_z$-axes, which ensures the validity of our discussions on the nodal structure. Also, as for the spin fluctuations in UTe₂, while NMR and $\mu$SR studies suggest the presence of ferromagnetic fluctuations [27,28], recent neutron scattering measurements show only antiferromagnetic fluctuations and no ferromagnetic ones [29–31]. A possible origin of this descrepancy is the measurement time-scale because the NMR study suggests that the ferromagnetic fluctuations are as slow as the order of kHz, which is too slow to be detected by neutron scatterings. While the magnetic fluctuations are still highly controversial, the ferromagnetic fluctuations may play an important role to induce the $B_{3u} + iA_u$ state, since the magnetic field along $a$-axis makes a degenerate state of the $B_{3u}$ and $A_u$ state. Thus, our experimental results promote further studies on the pairing mechanism in UTe₂.

Finally, we note that our conclusion of the $B_{3u} + iA_u$ pairing state is apparently different from the recent report of field angle-resolved specific heat measurements suggesting the point nodes only on the $k_x$-axis [8]. However, our anisotropic measurements in the Meissner state probe the ground state in the zero-field limit, while the application of strong magnetic field can change the superconducting symmetry [14,21,22,25]. How the $B_{3u} + iA_u$ state found here changes as a function of field also deserves further investigations.

## Summary and perspectives

To sum up, we have found that the frequency shifts of TDO circuit follow $T^n$ dependence with $n$ nearly equal to 2 or less than 2 regardless of sample quality or magnetic field direction. This rules out any of single-component odd-parity states, and thus indicates a multi-component order parameter. The anisotropic penetration depth analysis reveals the presence of multiple point nodes near the $k_y$- and $k_z$-axes, which is most consistent with the chiral $B_{3u} + iA_u$ superconducting state in UTe$_2$. The presence of TRSB components splits the point node to multiple point nodes away from high-symmetry axes, and in analogy to topological Weyl points in Weyl semimetals, these nodes are expected to create surface arcs of zero-energy Majorana quasiparticles states[34,39]. Thus UTe$_2$ is an ideal platform to investigate chiral superconductivity and its related topological physics. In particular, the positions of multiple point nodes (Weyl points) in the bulk studied here are fundamentally important to determine the topological properties of surface states.

## Methods

Single crystals of UTe$_2$ #B1, #C1, and #C2 were grown by the chemical vapor transport method with iodine as the transport agent. Crystal #A1 is grown by the molten salt flux method using a mixture of KCl-NaCl as flux[15] and picked up from the same batch with the crystal showing the RRR about 1000. In both cases, a slightly uranium-rich composition was employed for the starting ratio of U to Te to avoid uranium deficiency[17]. The details of the characterizations of the crystals are shown in Supplementary Information V.

To obtain anisotropic components of penetration depth $\Delta\lambda_a(T)$, $\Delta\lambda_b(T)$, and $\Delta\lambda_c(T)$ data separately, we have performed high-precision measurements of ac magnetic susceptibility shift $\Delta\chi(T) \equiv \chi(T) - \chi(0)$ using a tunnel diode oscillator technique operated at 13.8 MHz with weak ac magnetic field $H_\omega$ along the three crystallographic axes[40,41]. The ac magnetic field $H_\omega$ induced by the coil of the oscillator is the order of $\mu$T, which is much lower than the lower critical field of the order of mT in UTe$_2$[20]. In this technique, the frequency shift of the oscillator $\Delta f(T) \equiv f(T) - f(0)$ is proportional to $\Delta\chi(T)$.

Specific heat capacity was measured by a long relaxation method in a $^3$He cryostat or a dilution refrigerator, where a Cernox resistor is used as a thermometer, a heater and a sample stage. The bare chip is suspended from the cold stage in order that it has weak thermal link to the cold stage, and electrical connection for the sensor reading. The samples are mounted on the bare chip using Apiezon N grease. The specific heat of the crystals is obtained by subtracting the heat capacity of bare chip and grease from the total data.

## Data availability

The data that support the findings of this study are available within the paper and its Supplementary Information. Source data are provided with this paper.

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

## Acknowledgements

We thank S. Fujimoto, J. Ishizuka, T. Matsushita and Y. Yanase for fruitful discussions, and N. Abe, and T. Arima, M. Konczykowski, and Y. Tokunaga for technical supports. This work was supported by Grants-in-Aid for Scientific Research (KAKENHI) (Nos. JP22H00105, JP21J10737, JP21H01793, JP21KK0242, JP20H02600, JP19H00649, JP18H05227, JP16KK0106), Grant-in-Aid for Scientific Research on innovative areas "Quantum Liquid Crystals" (No. JP19H05824), Grant-in-Aid for Scientific Research for Transformative Research Areas (A) "Condensed Conjugation" (No. JP20H05869) from Japan Society for the Promotion of Science (JSPS), and CREST (No. JPMJCR19T5) from Japan Science and Technology (JST).

## Author contributions

K.H. and T.S. conceived the project. K. Ishihara, M.R., M.K., K.H., and T.S. performed magnetic penetration depth measurements and analyzes the results. K. Imamura and Y.M. carried out specific heat measurements. H.S., P.O., Y.T., and Y.H. synthesized and characterized single crystals of $UTe_2$. K. Ishihara, K.H., and T.S. prepared the manuscript with inputs from Y.H. All authors discussed the experimental results.

## Competing interests

The authors declare no competing interests.
