## [Peer Review File · Nature Communications]

Chiral superconductivity in UTe₂ probed by anisotropic low-energy excitationsEditorial Note: This manuscript has been previously reviewed at another journal that is not operating a transparent peer review scheme. This document only contains reviewer comments and rebuttal letters for versions considered at *Nature Communications*.

Reviewer #3 (Remarks to the Author):

I have read the response of the authors to previous comments and conclude that they have indeed adequately addressed all the issues raised. In particular, the new low temperature specific heat data do indeed place strong constraints on possible gap symmetries when combined with the penetration depth data. I do not have any further comments, so I am happy to recommend that the paper is published in Nature Comms.

Reviewer #4 (Remarks to the Author):

The editor asks for my opinion on the answers given to reviewer 2 by the authors. This work consists of very high quality measurements of magnetic penetration depth on crystals of the best qualities currently available on this fascinating material UTe_2 . Whatever the direction probed, the temperature dependence of the magnetic penetration depth follows a polynomial law close to T^2 . The authors show that these experimental facts are not compatible with a superconducting order parameter with a single component of a triplet spin state. They explain their result by a chiral $B_{3u}+iA_u$ state.

Both referees agree that the quality of the data is very high, and that the contribution of the better quality crystal data is remarkable. Moreover, they agree that this work should be published somewhere. Finally, they agree that the authors' interpretation is possible, but clearly not definitive. In particular, reviewer no. 2 raised several points concerning:

- i) How the T^2 law in all directions can be obtained in the case of $B_{3u}+iA_u$ state, whatever the rate of impurities scattering?
- ii) he asks clarification of the origin of the double transition in specific heat on lower quality samples
- iii) Estimation of the absolute penetration length from H_{c2}
- iv) Influence of multigap superconductivity

I agree with the referees in particular that the proposed scenario is consistent with the experimental data, but that it is not the only possible scenario. This scenario is surprising since the compatibility with the experimental data need a particular set of specific parameters of c_i characterising the mixture of $B_{3u}+iA_u$ and several strong hypothesis. On other hand, the authors have followed the referees's indications and have taken the necessary care in interpreting their measurements. I guess that the manuscript has been strongly improved by the authors, thanks to the referee's criticisms. It is now mainly a question of writing style, and the reader will be able to do is own opinion on the pertinence of the model proposed by the authors.

Given the quality of the experimental data on the one hand, and the interesting analysis proposed on another hand, this manuscript requires all the necessary qualities to be published. For my point of view, it meets the criteria of the editorial policy of Nature comm.

I suggest a small modification : please clarify the Fig S12b by specifying the value of Φ_N (Is it 30° , as in the case of FigS12a?)

Reviewer #3 (Remarks to the Author):

I have read the response of the authors to previous comments and conclude that they have indeed adequately addressed all the issues raised. In particular, the new low temperature specific heat data do indeed place strong constraints on possible gap symmetries when combined with the penetration depth data. I do not have any further comments, so I am happy to recommend that the paper is published in Nature Comms.

We thank the reviewer for the recommendation for the publication of our manuscript in *Nature Communications*.

Reviewer #4 (Remarks to the Author):

The editor asks for my opinion on the answers given to reviewer 2 by the authors.

This work consists of very high quality measurements of magnetic penetration depth on crystals of the best qualities currently available on this fascinating material UTe₂. Whatever the direction probed, the temperature dependence of the magnetic penetration depth follows a polynomial law close to T^2 . The authors show that these experimental facts are not compatible with a superconducting order parameter with a single component of a triplet spin state. They explain their result by a chiral $B_{3u}+iA_u$ state.

Both referees agree that the quality of the data is very high, and that the contribution of the better quality crystal data is remarkable. Moreover, they agree that this work should be published somewhere. Finally, they agree that the authors' interpretation is possible, but clearly not definitive. In particular, reviewer no. 2 raised several points concerning:

- i) How the T^2 law in all directions can be obtained in the case of $B_{3u}+iA_u$ state, whatever the rate of impurities scattering?
- ii) he asks clarification of the origin of the double transition in specific heat on lower quality samples
- iii) Estimation of the absolute penetration length from H_{c2}
- iv) Influence of multigap superconductivity

I agree with the referees in particular that the proposed scenario is consistent with the experimental data, but that it is not the only possible scenario. This scenario is surprising

since the compatibility with the experimental data need a particular set of specific parameters of c_i characterising the mixture of B3u+iA1u and several strong hypothesis. On other hand, the authors have followed the referees's indications and have taken the necessary care in interpreting their measurements. I guess that the manuscript has been strongly improved by the authors, thanks to the referee's criticisms. It is now mainly a question of writing style, and the reader will be able to do is own opinion on the pertinence of the model proposed by the authors.

Given the quality of the experimental data on the one hand, and the interesting analysis proposed on another hand, this manuscript requires all the necessary qualities to be published. For my point of view, it meets the criteria of the editorial policy of Nature comm.

We thank the reviewer for the thoughtful consideration of our manuscript and for stating that our manuscript meets the criteria of the editorial policy of *Nature Communications*.

I suggest a small modification : please clarify the Fig S12b by specifying the value of Φ_N (Is it 30° , as in the case of FigS12a?)

We appreciate the valuable comment. As the reviewer pointed out, the exponent values in Fig. S12b are calculated with the fixed value of $\phi_n = 30^\circ$. We have clarified this point in the revised manuscript.